# Scanning Electron Microscope Examination as an Alternative to Classical Microbiology in the Diagnostics of Catheter-Related Sepsis?

**DOI:** 10.3390/ijerph20065028

**Published:** 2023-03-13

**Authors:** Anna Kluzik, Hanna Tomczak, Marek Nowicki, Małgorzata Grześkowiak, Krzysztof Kusza

**Affiliations:** 1Department of Anaesthesiology, Intensive Therapy and Pain Treatment, Poznan University of Medical Sciences, 60-355 Poznan, Poland; k-kusza@ump.edu.pl; 2Department of Teaching Anaesthesiology and Intensive Therapy, Poznan University of Medical Sciences, 61-861 Poznan, Poland; mgrzesko@ump.edu.pl; 3Central Microbiology Laboratory, Clinical Hospital at the Poznan University of Medical Sciences, 60-355 Poznan, Poland; tomczak.hanna@spsk2.pl; 4Center for Advanced Technology, Adam Mickiewicz University, 61-614 Poznan, Poland; marek.nowicki@amu.edu.pl

**Keywords:** scanning electron microscope, SEM, catheter-related sepsis, CRBSI, intensive care unit, clinical microbiology

## Abstract

Central venous catheters are essential elements enabling the treatment of intensive care unit patients. However, these catheters are sometimes colonised by both bacteria and fungi, and thus, they may become a potential source of systemic infections—catheter-related bloodstream infections (CRBSI). The identification of the pathogen responsible for CRBSI is a time-consuming process. At the same time, the relationship between the quick identification of the pathogen and the implementation of targeted antibiotic therapy is of key importance for controlling the clinical symptoms of sepsis and septic shock in the patient. Quick diagnosis is of key importance to reduce morbidity and mortality in this group of patients. In our study, we attempted to create a catalogue of images of the most commonly cultured pathogens responsible for CRBSI. An FEI Quanta 250 FEG Scanning Electron Microscope (SEM) was used for measurements. SEM images obtained during the analysis were included in this study. Images of SEM are three-dimensional and comparable to the images seen with the human eye and are a tool used for research and measurement whenever it is necessary to analyse the state of the surface and assess its morphology. The method described in our study will not replace the current procedures recognised as the gold standard, i.e., pathogen culturing, determination of the count of microorganisms (CFU –colony forming units), and assessment of drug sensitivity. However, in some cases, the solution proposed in our study may aid the diagnosis of patients with suspected catheter-related bloodstream infections leading to sepsis and septic shock.

## 1. Introduction

The need to search for new methods enabling faster microbiological diagnostics, especially in critically ill patients, resulted in cooperation between the Poznan University of Medical Sciences and the Poznan University of Technology. Due to the development of electron microscopy techniques, they can currently be used in microbiological diagnostics, especially in vascular catheter-related infections. Central venous catheters are essential elements enabling the treatment of ICU (intensive care unit) patients, which involves drug infusion, parenteral nutrition, haemodynamic monitoring, and blood sampling for tests. However, these catheters are sometimes colonised by both bacteria and fungi, and thus, they may become a potential source of systemic infections. Paradoxically, progress in medicine, especially in organ transplantation, cancer treatment, extension of the survival time of patients undergoing immunosuppressive treatment as well as patients with impaired immunity, and the introduction of more invasive diagnostic and therapeutic methods, pose the risk of a greater number of complications, including those caused by catheter-related infections.

The incidence of catheter-related bloodstream infections (CRBSIs) ranges from 0.1 to 22.7/1000 catheter days. The aetiological factors responsible for these infections in ICU patients include Gram-positive (40–70.9%) and Gram-negative bacteria (1.8–77.8%) as well as yeasts. Infections are most often caused by Gram-positive bacteria such as *Staphylococcus aureus* and *Staphylococcus epidermidis* as well as other coagulase-negative staphylococci. The following Gram-negative bacilli are most often responsible for catheter-related sepsis: *Pseudomonas aeruginosa*, *Acinetobacter baumannii*, *E. coli*, *Klebsiella pneumoniae*, and *Enterobacter cloacae*. Due to their multi-drug resistance, their importance in the epidemiology of catheter-related infections is constantly growing. It significantly increases the morbidity and mortality of patients due to sepsis and septic shock [1,2,3].

Currently, the following methods are used to confirm CRBSI: catheter tip culture (CTC) with a value equal to or greater than 15 CFU (colony-forming units) and the culturing of the same pathogen from peripheral blood, RQC (ratio of quantitative culture), i.e., a comparison of the growth of pathogens from the blood collected simultaneously from a potentially infected central venous catheter and peripheral blood, and DTP (differential time to positivity), i.e., a comparison of the growth time of the pathogen from the blood collected simultaneously from a potentially infected central venous catheter and peripheral blood. A catheter-related infection is confirmed when the same pathogen is cultured from both biological materials and when the growth time of the CTC pathogen is at least 2 h ahead of the growth of the peripheral blood pathogen. However, these methods involve some technical difficulties and the waiting time for the final result is extended. It is necessary to remove the catheter from the patient’s vascular bed for a catheter tip culture (CTC). According to the recommendations of the Infectious Diseases Society of America (IDSA) for multi-lumen catheters, a catheter-related infection is confirmed when blood has been collected from all catheter channels and a pathogen culture exceeds 100 CFU/mL. Regardless of the method chosen, the identification of the pathogen responsible for catheter-related infection is a time-consuming process. At the same time, the relationship between the quick identification of the pathogen and the implementation of targeted antibiotic therapy is of key importance for controlling the clinical symptoms of sepsis and septic shock in the patient. Therefore, it is necessary to search for new diagnostic solutions which will significantly accelerate the identification of the pathogen responsible for catheter-related sepsis in order to apply targeted antibiotic therapy. Quick diagnosis is of key importance to reduce morbidity and mortality in this group of patients [1,2,3,4].

## 2. Research Method and Group

At this stage of the research, a catalogue of the most commonly cultured pathogens responsible for the incidence of catheter-related sepsis in our centre was created. Reference strains were used as representatives of individual species of pathogens. A total of 10 microlitres of the microbial suspension ware added to test tubes with sterile brain heart infusion (BHI) broth. Next, two sterile fragments (1 cm long) of the central venous catheter were added to each test tube. After the assumed cultivation time, i.e., after 24 and 48 h, respectively, the central venous catheters were aseptically removed and sent to the microscopy laboratory for analysis.

An FEI Quanta 250 FEG Scanning Electron Microscope (SEM) was used for measurements. The analysis was conducted in the low vacuum mode (70–80 Pa in the microscope chamber), without any special preparation and without covering the material with conductive layers. An accelerating voltage of 5 kV was used. The samples were fixed with carbon tape on standard SEM tables. The wet samples were analysed after ethanol evaporation within 10–30 s.

## 3. Results

SEM images obtained during the analysis were included in this study. The images obtained after 24 and 48 h of culturing did not differ from each other. This proved the rapid colonization of central catheters in the presence of pathogens in the environment where the central catheter was located. No biofilm was observed on any of the examined catheter fragments. The SEM test was conducted shortly after the material delivery, i.e., in about 15 min. Images of the reference strains from the scanning force microscope are shown as Figure 1. Figure 1 includes representatives of individual pathogens responsible for catheter sepsis characteristic of catheter-related-sepsis in our centre. Images show characteristic bacterial cells: cocci, rod-shaped, and the image corresponds to at least one of the following features: cell division or a characteristic spatial arrangement, cell clusters compatible with a given microcolony (staphylococci, bunches streptococci). Yeast-like fungi were much larger than bacteria.

## 4. Discussion

Scanning electron microscope images are three-dimensional and comparable to the images seen with the human eye. Scanning electron microscopy (SEM) is a tool used for research and measurement whenever it is necessary to analyse the state of the surface and assess its morphology. It is used both for living and inanimate particles. Due to high resolution and magnification, SEM enables a detailed analysis of various mechanisms occurring in the world of biology, medicine, ecology, biotechnology, and microbiology. Moreover, SEM measurement is a fairly quick procedure. It takes about 15 min to obtain an image after the delivery of material for tests. Due to a very large range of available magnifications (100–1,000,000×), SEM enables quick identification of the place with microorganisms even if they cover only a small percentage of the catheter surface (usually < 1%). SEM images have depth, which facilitates the assessment of examined structures [5,6]. These properties were used to assess the stages of the replication cycle of particles of Ebola virus (EBOV) and cells infected with it at the ultrastructural level [7]. SEM was also used to confirm a chronic *Chlamydia* infection and colonisation. These bacteria are responsible for the severity of chronic diseases such as atherosclerosis as well as emphysema [8]. SEM is also used for the assessment of the shape, size, and location of microorganisms in biofilm as well as the stages of biofilm formation, i.e., bacterial interactions and the production of extracellular polymer substances [6]. SEM was applied to assess the colonisation of exposed orbital implants by pathogens. The examination showed that the bacteria which usually constitute the conjunctival microflora had high capacity for adhesion and biofilm formation. The SEM image of the cement structure of the periodontium showed that this substance covering the tooth root provided anchorage to periodontal fibres. Both studies confirmed the presence of pathogens [9,10].

Our research was conducted on a group of ICU patients with suspected catheter-related blood infection. It showed that SEM could aid traditional microbiological tests by shortening the time of diagnosis before the implementation of appropriate antibiotic therapy [11]. There were also studies on the effect of antibiotics on biofilm cells and the behaviour of various biomedical materials under the influence of antibiotics. This means that SEM could be used to understand the effectiveness of antimicrobial therapies [12]. Researchers also used SEM to assess thrombogenicity and the presence of traits indicating the colonisation of central venous catheters by comparing catheters with and without the treated surface [13]. SEM was used to assess the colonisation of epidural catheters by comparing the images of their individual fragments with the results of microbial cultures. The SEM images suggested that the skin was the main source of infections. Additionally, the formation of a fibrin network in the lumen of the catheter with a low level of bacterial colonisation was confirmed. The network negatively affected the patency of the catheter [14].

Due to preliminary information on the appearance of microorganisms colonising the central venous catheter in a patient with suspected catheter-related infection and due to the data based on microbiological mapping, it is possible to quickly initiate targeted antibiotic therapy. Microbiological mapping is a necessary element to make therapeutic decisions, create and modify the hospital antibiotic prescription, shape the hospital antibiotic policy, and control infections related to medical care. At the current stage of research, it was not possible to prove that there were characteristic traits of SEM images of microorganisms, indicating their resistance to individual antibiotics.

## 5. Conclusions

It usually takes 3–4 days to obtain the result of the test confirming the presence of the pathogen in the blood of a patient with suspected catheter-related sepsis. The waiting time for a catheter tip culture is the same. This time is necessary for a culture conducted in a laboratory. If a patient is suspected of CRBSI and requires catheter removal, this time may be shortened by assessing the catheter in a microscopy laboratory. Such cooperation is usually possible only in large clinical centres. Our research team tried to create a catalogue of reference pathogens responsible for CRBSI which most often occur in our centre, in order to enable quick initial diagnosis of bloodstream infections. The ability to visualize pathogens on the central catheter and compare them with the database-catalogue enables the shortening of incubation, i.e., 24 h for bacteria and up to 48 h for candida.

It is neither an ideal method nor does it fully meet the diagnostic needs related to CRBSI. The method does not enable the identification of a specific pathogen species. However, it enables the identification of certain groups of pathogens, i.e., cocci (forming clusters, chains), bacilli or yeast-like fungi, which is of great practical importance. Due to quick initial information on the type of pathogen and data from the microbiological mapping of a particular centre, it is possible to initiate appropriate antibiotic therapy or administer an antifungal drug, according to the scope of action.

It takes time to culture pathogens from the blood. The SEM test shortens the process of initial microbiological diagnosis. Due to cooperation with a centre using a scanning electron microscope (SEM), it is possible to preliminarily confirm the presence of microorganisms and a specific group of pathogens on the surface of catheters collected from patients with suspected catheter-related bloodstream infection.

Due to the preliminary catalogue-based identification of the type of pathogen shortly after the removal of a central venous catheter from the patient’s body, it is possible to accelerate therapeutic activities, i.e., implement antibiotic therapy against Gram-positive or Gram-negative pathogens and fungi.

### Disadvantages of the Method

Currently, the wide application of this method in the health care system is not possible due to logistical reasons (providing material for testing). It can be used only by large clinical centres cooperating with a microscopy laboratory.Currently, the diagnostic procedure still requires the culturing of microorganisms in order to prepare an antibiogram enabling the selection of a targeted antimicrobial drug.For the SEM analysis, the central venous catheter must first be removed. Bacterial or candida colonization in the lumen biofilm of central venous catheters is common (inside of the catheter must be always examined).The presence of bacteria or candida detected by SEM does not mean that it is the source of infection. Other causes of bacteraemia should be excluded.The central venous catheter may be covered with a biofilm, and multimicrobial bacteraemia with CRBSI is often observed, which prevents the visualisation of characteristic traits of the pathogen type.The method still requires further standardisation and extension of the catalogue with new pathogens. To make the method more useful, a large bank of SEM images should be created.Correct matching of the new SEM images to the images from the image bank would require the creation of algorithms/models that would enable quantitative assessment and, thus, support decisions. Manually adjusting images can cause errors.

The method described in our study will not replace the current procedures recognised as the gold standard, i.e., pathogen culturing, determination of the count of microorganisms (CFU–colony forming units), and assessment of drug sensitivity. However, in some cases, the solution proposed in our study may aid the diagnosis of patients with suspected catheter-related bloodstream infections leading to sepsis and septic shock. Above all, it may significantly accelerate the initial selection of the antibiotic, especially when the result of the microbial culture is negative, whereas the clinical condition of the patient clearly indicates sepsis or septic shock. Infections in the ICU are usually caused by multi-drug-resistant pathogens. Therefore, broad-spectrum antibiotics are usually administered to patients as drugs of last resort. However, it is important to know whether to use drugs against cocci (usually Gram-positive), bacilli (usually Gram-negative), or fungi, which is particularly important due to the high mortality in the course of systemic mycosis. It would be interesting to investigate, in further research, whether and when the SEM test will be specific and sensitive enough to replace the antibiogram, i.e., whether it will be possible to visualise the resistance characteristics of the pathogen.

The implementation of microbiological diagnostics requiring the presented SEM technology may be of common importance in healthcare organization in the future. Currently, in our opinion, this is not possible for logistical and technological reasons.

In our research, we used the microscope laboratory of the University of Technology, to which hospital facilities currently do not have permanent access. Establishing cooperation between large university centres can improve diagnostics in many areas—in this case, it will speed up the confirmation of the presence of the pathogen and the implementation of empirical treatment.

Microbiological diagnostics requiring the presented SEM technology may be of common importance in healthcare organisations in the future. Currently, in our opinion, this is not possible for logistical and technological reasons.

## Figures and Tables

**Figure 1 ijerph-20-05028-f001:**
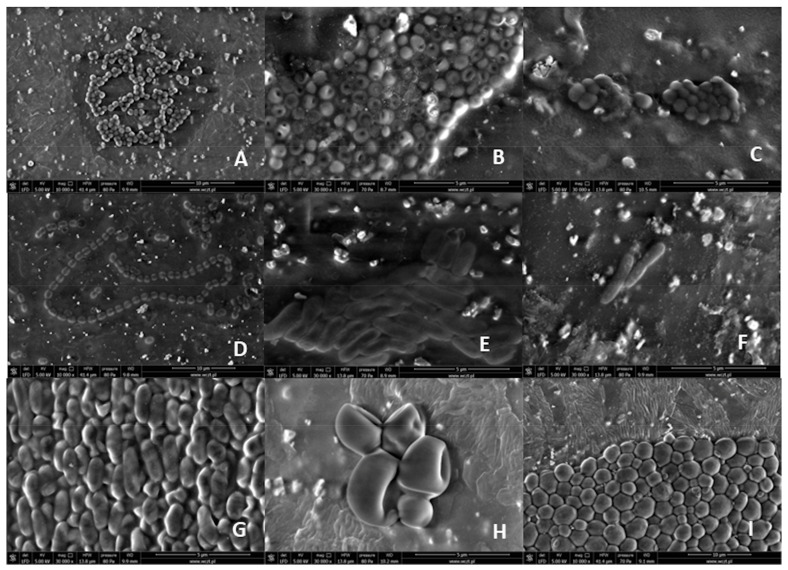
Scanning electron microscopy (SEM) images of reference pathogens: Gram-positive cocci: (**A**) *Staphylococcus aureus*, (**B**) *Staphylococcus epidermidis*; Gram-positive cocci—streptococci: (**C**) *Streptococcus parasanguinis*; Gram-positive enterococci: (**D**) *Enterococcus fecalis*; Gram-negative bacilli (**E**) *Klebsiella pneumonie*, (**F**) *Escherichia coli*, (**G**) *Pseudomonas aeruginosa*; yeast-like fungi of the *Candida* genus: (**H**) *Candida dubliniensis*, (**I**) *Candida albicans*.

## Data Availability

All data supporting the reported results are presented in the manuscript.

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
