# Peer review of "Scanning Electron Microscope Examination as an Alternative to Classical Microbiology in the Diagnostics of Catheter-Related Sepsis?"

_ijerph, 2023, doi:10.3390/ijerph20065028_

Round 1
Reviewer 1 Report
It is a good initiative and can be easily seen that the author group has done extensive work to have proposed this method of identifying groups of pathogens as weather cocci (forming clus- 168 ters, chains), bacilli or yeast-like fungi with a quick run.
Author Response
Dear Reviewer,
thank you for your comments and recommendations. Suggested changes have been made in the text.
Kind regards
Authors
Reviewer 2 Report
Dear Authors,
Thank you for submitting this interesting paper. There are some room for improvement. In list of disadvantages of SEM methods, consider the following be listed:
1). it is very common that bacterial or candida colonization within the biofilm of the lumen of central venous catheters. The presence of bacteria or candida detected on your SEM method does not absolutely mean it is a CRBSI.
2). For SEM method, the central venous catheter must be removed first for analysis with SEM technique. The SEM technique does not prevent unnecessary central venous line removal if not CRBSI as the central line has already been out for SEM analysis.
3). It is common to see polymicrobial bacteremia from CRBSI. Do not know how SEM can differentiate between among different bacteria.
4). Sometimes, positive or negative SEM test may not matter. For example, if the Staph aureus bacteremia from a different source but the patient has a central venous catheter at the same time of bacteremia. Then, we usually do recommend removal of central venous catheter because we have concern of secondary seeding of Staph aureus to the central lines.
5). Any bacteria can cause CRBSI - for example, Stenotrophomonas maltophilia, Nocardia species, non-tuberculous Mycobacteria, Leuconostoc species, and etc. You would need to have a huge Image bank to match those organisms.
6). I would like to know how to match SEM images from catheter and stored images. Is it automation or manually? If manually, there would be operator discrepancy.
Those are my comments and recommendations.
Thank you.
Author Response
Dear Reviewer,
thank you for your comments and recommendations. Suggested changes have been made in the text. The conclusions and especially limitations of the method are described in more detail.
Response to 1: indeed, pathogen colonization often occurs in the lumen of a central catheter. This is a very valid point. Our research in the first stage also included SEM viewing of the lumen of the catheters (Reference No. 12). In the second stage, the catheters were also examined from the lumen after 24 and 48 hours. Information is provided in the text.
Response to 2: Suspicion of catheter infection (on the basis of clinical symptoms, increase in inflammatory parameters, length of catheter stay) still forces removal of the catheter as a source of infection and insertion of a new catheter.
Response to 3: Evaluation of catheters by SEM is not a perfect technique. The presence of biofilm covering pathogens and the presence of multiple pathogens colonizing the catheter may make this method difficult to use.
Response to 4: I agree, sometimes a positive or negative SEM test result may not matter.
Response to 5: I agree, the method still requires further standardisation and extension of the catalogue with new pathogens. To make the method more useful, a large bank of SEM images should be created.
Response to 6: Correct matching of the new SEM images to the images from the image bank would require the creation of algorithms/models (as programs supporting the work of radiologists) that would enable quantitative assessment and thus support decisions. Manually adjusting images can cause errors.
Authors
Reviewer 3 Report
This study aimed to use n FEI Quanta 250 FEG Scanning Electron Microscope (SEM) create a catalog of images of the most commonly cultured pathogens responsible for catheter-related bloodstream infection (CRBSI). However, I had serious concern about the study design and findings. Regarding the study method, there is no information about the case identification and how the pathogens collected. About the results, the description was too limited. In addition, the discussion is too lengthy and too over to interpret their findings. Finally, I do not think these limited findings can support their conclusion. Overall, it cannot be published as an orignial study.
Author Response
Dear Reviewer,
Thank you for your comments and recommendations.
Response to 1: In this part of the study, we used reference strains from the microbiology laboratory of our hospital as representatives of particular species of pathogens ("Reference strains were used as representatives of individual species of pathogens. 10 microliters of the microbial suspension were added to test tubes with sterile Brain Heart Infusion (BHI) broth. Next, two sterile fragments (1 cm long) of the central venous catheter were added to each test tube. After the assumed cultivation time, i.e., after 24 and 48 hours, respectively, the central venous catheters were aseptically removed and sent to the microscopy laboratory for analysis"). In the first part of the study, catheters removed from patients were evaluated (Reference No. 12)
.
Response to 2: The description of the results is relatively short because the purpose of this part of the study was to image pathogens of known origin using SEM to create a catalog of SEM images. The SEM pictures catalog is still under construction - it should contain many more pathogens and include representatives of particular groups. The currently depicted reference strains (9) are characteristic of our unit.
Response to 3: The results and conclusions have been extended.
Kind regards
Authors
Round 2
Reviewer 2 Report
Dear Authors,
Thank you for your editing and re-submission.
The method you used in this study is likely not get applied in real practice. Currently, bacterial or fungal identification methods have evolved significantly with shorter duration of accurate identification of causal pathogens from blood cultures or catheter tip cultures.
Author Response
Dear Reviewer,
thank you for the comment. We agree that the method proposed by us can only be used by large clinical center cooperating with a microscopy laboratory. It is not method for general use. After achieve the full SEM images catalog of pathogens, the method proposed by us may have comparable or greater sensitivity and specificity than current diagnostic methods. But it should be proven. Moreover, it has an educational aspect and research purpose.
Kind regards
Authors
Reviewer 3 Report
The authors response well, so I have no more comment.
Author Response
Dear Reviewer,
thank you for your answer without a new comment.
Kind regards
Authors